# Central Serotonin Deficiency Impairs Recovery of Sensorimotor Abilities After Spinal Cord Injury in Rats

**DOI:** 10.3390/ijms26062761

**Published:** 2025-03-19

**Authors:** Yuri I. Sysoev, Polina Y. Shkorbatova, Veronika A. Prikhodko, Daria S. Kalinina, Elena Y. Bazhenova, Sergey V. Okovityi, Michael Bader, Natalia Alenina, Raul R. Gainetdinov, Pavel E. Musienko

**Affiliations:** 1Department of Neuroscience, Sirius University of Science and Technology, Sirius 353340, Russia; sysoevyi@infran.ru (Y.I.S.);; 2Pavlov Institute of Physiology of the RAS, Saint Petersburg 199034, Russia; polinavet@yandex.ru (P.Y.S.); bazhelen@mail.ru (E.Y.B.); 3Institute of Translational Biomedicine, Saint Petersburg State University, 7–9 Universitetskaya Emb., Saint Petersburg 199034, Russia; gainetdinov.raul@gmail.com; 4Department of Pharmacology and Clinical Pharmacology, Saint Petersburg State Chemical and Pharmaceutical University, Saint Petersburg 197022, Russia; veronika.prihodko@pharminnotech.com (V.A.P.); okovityy@mail.ru (S.V.O.); 5Sechenov Institute of Evolutionary Physiology and Biochemistry of the RAS, Saint Petersburg 194223, Russia; 6Max Delbrück Center for Molecular Medicine in the Helmholtz Association, 13125 Berlin, Germany; mbader@mdc-berlin.de (M.B.); natalenina2014@gmail.com (N.A.); 7Federal Center of Brain Research and Neurotechnologies, Moscow 199330, Russia; 8Life Improvement by Future Technologies Center, Moscow 143025, Russia

**Keywords:** serotonin, TPH2, spinal cord injury, lateral hemisection, sensorimotor function, tapered beam walking test, ladder walking test, static rod, compound muscle action potential

## Abstract

Spinal cord injury (SCI) affects millions of people worldwide. One of the main challenges of rehabilitation strategies is re-training and enhancing the plasticity of the spinal circuitry that was preserved or rebuilt after the injury. The serotonergic system appears to be crucial in these processes, since recent studies have reported the capability of serotonergic (5-HT) axons for axonal sprouting and regeneration in response to central nervous system (CNS) trauma or neurodegeneration. We took advantage of tryptophan hydroxylase 2 knockout (TPH2 KO) rats, lacking serotonin specifically in the brain and spinal cord, to study the role of the serotonergic system in the recovery of sensorimotor function after SCI. In the present work, we compared the rate of sensorimotor recovery of TPH2 KO and wild-type (WT) female rats after SCI (lateral hemisection at the T8 spinal level). SCI caused severe motor impairments in the ipsilateral left hindlimb, the most pronounced in the first week after the hemisection with gradual functional recovery during the following 3 weeks. The results demonstrate that TPH2 KO rats have less potential to recover motor functions since the degree of sensorimotor deficit in the tapered beam walking test (TBW) and ladder walking test (LW) was significantly higher in the TPH2 KO group in comparison to the WT animals in the 3rd and 4th weeks after SCI. The recovery dynamics of the hindlimb muscle tone and voluntary movements was in agreement with the restoration of motor performance in TBW and LW. Compound muscle action potential analysis in the gastrocnemius (GM) and tibialis (TA) muscles of both hindlimbs after electrical stimulation of the sciatic nerve or lumbar region (L5–L6) of the spinal cord indicated slower recovery of sensorimotor pathways in the TPH2 KO group versus their WT counterparts. In general, the observed results confirm the significance of central serotonergic mechanisms in the recovery of sensorimotor functions in rats and the relevance of the TPH2 KO rat model in studying the role of the 5-HT system in neurorehabilitation.

## 1. Introduction

Spinal cord injury (SCI) is one of the most devastating events, affecting millions of people worldwide [1]. It is often associated with life-threatening complications and has dramatic consequences for the psychosocial and economic well-being of patients and their caregivers. To date, several therapeutic approaches for the management of SCI patients have been proposed [2]. Among them, one of the most promising is epidural electrical stimulation with skilled motor training [3]. In individuals with SCI, this has led to independent standing and stepping with improvements in autonomic functions [4]. Although the abovementioned strategies have some beneficial effects, there is no effective treatment to promote axonal regeneration with full functional recovery.

A series of studies has suggested that the serotonergic system possesses specific regenerative abilities for active axonal sprouting and further regeneration after SCI. Of note, in rodent models of SCI immediately rostral to the lesion site, an increase in serotonergic (5-HT) axon density was seen, although 5-HT axonal degeneration caudal to the lesion site has also been observed [5]. Elevated 5-HT neurotransmission (5-hydroxytryptophan administration) promoted better motor rehabilitation and improved recovery of pain behavior both in rat [6,7] and cat [8] models of SCI. Antidepressant treatment [9], electrical stimulation of raphe magnus [10] or intrathecal 5-HT administration [11] had similar positive effects on motor function recovery. In summary, these findings indicate that induction of serotonergic transmission after SCI has beneficial effects on functional recovery. Undoubtedly, all the abovementioned studies clearly demonstrate the significant role of 5-HT in neurorehabilitation.

In the present study, we aimed at clarifying if the lack of serotonergic neurotransmission affects the recovery process after SCI. The use of animal models in which the synthesis of 5-HT is disrupted only in the neurons within the central nervous system can significantly expand our understanding of the serotonergic mechanisms of neuronal recovery after SCI. We took advantage of genetically modified rats, lacking tryptophan hydroxylase 2 (TPH2) [12], the rate-limiting enzyme of 5-HT synthesis in the central nervous system [13] (Figure 1A,B). We compared the recovery of sensorimotor functions in TPH2-deficient (TPH2 KO) and wild-type (WT) rats after lateral hemisection of the spinal cord. We found out that central serotonin plays an essential role in the recovery of sensorimotor abilities after spinal cord injury.

## 2. Results

### 2.1. TPH2 Deficiency Results in Serotonin Depletion in Rat Spinal Cord

We first evaluated whether TPH2-deficiency affects serotonin levels in the spinal cord. We measured the levels of serotonin and its degradation product, 5-hydroxyindoleacetic acid (5-HIAA), in the cervical, thoracic and lumbar areas of the spinal cord. As expected, both metabolites were almost undetectable in TPH2 KO rats (Figure 2). Furthermore, in WT rats, a clear difference in serotonin content was observed between different spinal cord regions, with the highest levels in the lumbar areas, corresponding to the previously reported serotonin distribution throughout the spinal cord [14].

### 2.2. A Lack of Central Serotonin Affects the Recovery Dynamics of Sensorimotor Capacities After SCI

We next assessed the motor performance in TPH2 KO and WT rats at basal conditions. In the present study, we did not observe any differences in motor performance between uninjured WT and TPH2 KO rats using the chosen testing protocol. A week before SCI, the results of both groups were similar in TBW (Figure 3A), LW (Figure 3B) and static rod (Figure 3C) tests. SCI induced severe motor deficiency in the left ipsilateral hindlimb. These disturbances were the most pronounced in the first week after the hemisection, with gradual functional recovery during the following 3 weeks (Figure 3). Quantification of the lesion size indicated that there were no significant differences in the length of the lesion and spared tissue area between the WT and KO groups (Figure 4).

#### 2.2.1. Rat Behavior in Tapered Beam Walking Test After SCI

In the first week after SCI, the degree of sensorimotor deficit in the ipsilateral hindlimb was approximately 100% in both groups in the TBW test (*p* < 0.01 vs. own values before surgery in both groups by Tukey’s multiple comparisons test; RM one-way ANOVA: F = 55.51, *p* < 0.01 and F = 2695, *p* < 0.01 for WT and KO, respectively). The WT group demonstrated gradual recovery of motor function, which became significant (*p* = 0.0202, by Tukey’s multiple comparisons test) in the 4th week after the surgery. Unlike the WT group, TPH2 KO rats had less potential for motor function recovery and did not show any differences in degree of sensorimotor deficit between the first and last week after SCI. Eventually, the degree of sensorimotor deficit in TBW was significantly higher in the TPH2 KO group vs. WT in the 3rd and 4th weeks after the surgery (*p* = 0.0152 and *p* = 0.0022 by Mann–Whitney U-test, in the 3rd and 4th weeks, respectively) (Figure 3A, Appendix A).

#### 2.2.2. Rat Behavior in Ladder Walking Test After SCI

Unlike the TBW test, SCI induced a 100% sensorimotor deficit in the LW test only in TPH2 KO rats in the 1st week after the surgery. Initially after injury, half of WT animals had about 50% while the others had a 100% sensorimotor deficit. In both groups, we observed trends towards recovery of hindlimb function during the following 4 weeks, although this recovery was significant in the 4th week after injury only in the WT group. Similar to the TBW test, in the 3rd and 4th weeks, the degree of sensorimotor deficit of the ipsilateral hindlimb was significantly higher in the TPH2 KO group compared to WT animals (*p* = 0.0216 and *p* = 0.0022 by Mann–Whitney U-test, in the 3rd and 4th weeks, respectively) (Figure 3B, Appendix A).

#### 2.2.3. Rat Behavior in Static Rod After SCI

The static rod test was the most difficult sensorimotor test for both groups since neither WT nor TPH2 KO rats demonstrated significant recovery, even in the 4th week after the injury. Despite the absence of statistical significance between groups at any timepoint, it should be mentioned that three of the WT rats exhibited partial recovery of ipsilateral hindlimb function (from 100% of sensorimotor deficit in the 1st week to approximately 50% in the 3rd and 4th weeks), which, undoubtedly, could be considered as positive group dynamics (Figure 3C, Appendix A).

#### 2.2.4. Muscle Tone and Voluntary Movements

The recovery dynamics of the hindlimb muscle tone and voluntary movements (Figure 5) was in agreement with that of motor performance in the TBW, LT and static rod tests (Figure 3). In the WT group, on the 1st day after hemisection, the hindlimb muscle tone was “absent” (left, ipsilateral) or “weak” (right, contralateral). Recovery of the left hindlimb to the level of “good” muscle tone (score 4) occurred only 9 days after the injury, and on the 12th day the muscle tone was “strong” (normal; score 5), not changing any longer. The TPH2 KO rats demonstrated less recovery capacity in comparison with the WT group. Indeed, we observed full recovery of right and left hindlimb muscle tone only on the 12th and 20th days after hemisection, respectively (Figure 5A).

By the 6th day after injury, the capability for voluntary movement of the left hindlimb was observed in all WT rats. In general, in the TPH2 KO group, right hindlimb recovery was slower in comparison with the WT rats. Although one of TPH2 KO rats showed recovery on the 2nd day, the ability to conduct voluntary movement of the left limb, ipsilateral to hemisection, was achieved in all rats only on the 12th day after the operation (Figure 5B).

### 2.3. TPH2 Deficiency Slows the Recovery of Sensorimotor Pathways in Rats

Electrophysiological assessment was performed four weeks after surgery in the TPH2 KO and WT rats. Stimulation of both sciatic nerves triggered responses with 0.62 ± 0.01 ms and 0.58 ± 0.02 ms latency in the WT group and 0.62 ± 0.02 ms and 0.60 ± 0.02 ms in the KO group in the TA and GM muscles, respectively (Figure 6B–E). The shape and amplitude of the threshold, submaximal and maximal responses were almost the same in TA_L vs. TA_R and GM_L vs. GM_R in the WT group (Figure 6B,C). In contrast, in the TPH2 KO group, submaximal and maximal responses had lower amplitudes in TA_L and GM_L compared to the corresponding muscles on the right side (Figure 6D,E). In both groups, stimulation amplitude required for threshold responses did not differ significantly and was approximately 5 mA. In general, the recruitment curves of all recorded muscles were superimposed on each other in the WT group. Unlike that, in the KO group, recruitment curves of TA_L and GM_L reached a plateau much earlier compared to TA_R and GM_R (Figure 6H).

The latencies of TA and GM responses during stimulation of the L5–L6 spinal cord were approximately 2.5-fold longer in comparison with SN stimulation (~1.60 ms) (Figure 6F,G). Generally, the obtained results for stimulation of the L5–L6 spinal cord were similar to the case of sciatic nerve stimulation (Figure 6F,G). For example, the shape and amplitude of the threshold, submaximal and maximal responses were almost the same between corresponding muscles on the left and right sides in the WT group. In the TPH2 KO group, the amplitude of submaximal and maximal responses was lower in TA_L and GM_L vs. TA_R and GM_R, respectively. Both groups had similar threshold stimulation amplitudes (~3 mA). Unlike the WT group, where the recruitment curves were similar in all four muscles, in TPH2 rats the recruitment curves of TA_R and GM_R achieved higher amplitudes vs. TA_L and GM_L (Figure 6I).

Neither the latency nor stimuli required for maximum response of the CMAP differed significantly between animals of both groups 4 weeks after hemisection. It should be mentioned that the ratio of maximum left and right hindlimb muscle responses (TA_L vs. TA_R and GM_L vs. GM_R) in the TPH2 KO group was less in comparison with WT animals. This tendency was observed with stimulation of both the sciatic nerve and L5–L6 spinal cord, and there was a significant difference between the TPH2 KO and WT groups in the ratios of TA_L vs. TA_R responses for SN stimulation (*p* = 0.0286, by Mann–Whitney U-test) and GM_L/GM_R for L5–L6 spinal cord stimulation (*p* = 0.0286, by Mann–Whitney U-test) (Figure 6J).

## 3. Discussion

In the present study, we demonstrated the significant role of central 5-HT neurotransmission in the recovery of motor function in rats after SCI. The obtained results clearly indicate that serotonin-deficient rats have less potential to recover motor functions, as the degree of sensorimotor deficit in the TBW and LW tests was significantly higher in the TPH2 KO group versus WT rats in the 3rd and 4th weeks after surgery. The recovery dynamics of hindlimb muscle tone and voluntary movement was in agreement with the restoration of motor performance in the abovementioned tests. Also, the compound muscle action potential analysis in the GM and TA muscles of both hindlimbs after stimulation of the sciatic nerve or lumbar region of the spinal cord indicated slower recovery of sensorimotor function in the TPH2 KO group versus the WT group. Altogether, these findings reaffirm the important role of the serotonergic system in restorative processes after SCI [15,16] in rats and demonstrate the relevance of the TPH2 KO model in such studies.

Numerous studies have reported that increased 5-HT neurotransmission promotes functional recovery after SCI [8,10,11,17]. On the one hand, 5-HT agonists can activate circuits of the lumbar spinal cord, thereby promoting recovery of motor functions. For example, 5-HT application has been shown to induce locomotor-like discharge of the lumbar ventral roots in vitro in neonatal rat brainstem–spinal cord preparation [18]. In paraplegic rats, activation of 5-HT receptors by quipazine and 5-HT1A/7 receptors by 8-hydroxy-2-(di-n-propylamino)-tetralin (8-OH-DPAT) led to the production of well-coordinated weight-supported locomotion with a reduced need for exteroceptive stimulation [19]. A 5-HT2 receptor-dependent pathway processed hindlimb locomotor-like proprioception to facilitate hindlimb stepping in response to epidural spinal cord stimulation in a decerebrated cat model [20].

In general, it is widely accepted that injured neurons in the central nervous system do not undergo axonal regrowth. Surprisingly, numerous recent studies have indicated the ability of 5-HT axons to undergo axonal sprouting and regeneration in response to CNS insult or neurodegeneration [21,22]. In a study by Jin et al. [21], the transection of serotonin axons in the neocortex in adult mice was followed by regrowth from cut ends into and across the stab rift zone. The regrown axons released serotonin, and their regrowth correlated with functional recovery. Similar results were obtained for traumatic brain injury, where in adult female mice serotonergic axons were capable of regrowing into the distal zone to increase axonal density by 1 month after the injury [19]. Moreover, in a recent study by Hou et al. [23], implantation of serotonergic (5-HT+) neuron-enriched embryonic raphe nucleus-derived neural stem cells/progenitors (RN-NSCs) into a complete spinal cord transection lesion site in adult female rats led to projection of serotonergic circuits to the caudal autonomic regions. The reconstituted serotonergic regulation of sympathetic activity led to the improvement of hemodynamic parameters and mitigated autonomic dysreflexia. At the same time, the serotonergic system may play a key role in the regulation of cortical plasticity, including sprouting of neurites, dendritic remodeling or synaptogenesis. For example, chronic 5-HT pharmacotherapy after complete spinal cord transection in rats enhanced reorganization in the sensorimotor cortex. This reorganization correlated positively with improvements in BBB (Basso, Beattie and Bresnahan) behavior score [24]. Altogether, these findings clearly indicate the essential role of 5-HT in axonal regeneration within the CNS.

Today, rats provide an important mammalian model to evaluate treatment strategies and to explore the pathological basis of SCI. In general, from these studies, we have learned much of what is known about the pathological events that follow after SCI, typically summarized as “secondary injury” [2,24,25]. Despite the many similarities with humans and hence the advantages of the rat SCI model for translational medicine, some explicit differences also occur. First of all, humans are much larger than rats and have larger volumes of gray matter that need to be reinnervated after SCI. Additionally, in humans, the metabolic rate of secondary injury processes is slower in comparison with small laboratory animals such as rodents. Due to these differences, the time scales for the proposed therapeutic approaches would also differ between rats and patients in clinical practice. Furthermore, in humans, SCIs are often heterogeneous (which causes difficulties when selecting patients for clinical trials), unlike injuries in experimental groups of rats. Nevertheless, studies in rat SCI have suggested several treatment options, some of which have advanced to clinical trials [26].

To date, this is the first study where the capability for recovery of sensorimotor function after SCI was evaluated in TPH2 KO rats. Previous studies in TPH2-deficient rats have documented a significant increase in both gene and protein expression of the brain-derived neurotrophic factor (BDNF) in the prefrontal cortex of adult TPH2 KO rats in comparison to WT counterparts [27]. The authors suggested that such findings implied a potential compensatory mechanism in the brain to cope with the absence of the trophic contribution of serotonin. Similar changes were also shown for the hippocampus of TPH2 KO mice [28,29], which invokes a logical assumption that BDNF upregulation is not a feature of individual brain structures but of the CNS in general. BDNF is one of the best described neurotrophic factors that regulate neuronal survival and differentiation, and it functions in activity-dependent plasticity processes such as long-term potentiation, long-term depression (LTD) and learning memory [30]. Animal studies have shown its significant neuroprotective action in the rubro-, reticulo- and vestibulospinal tracts, as well as in the proprioceptive neurons of Clarke’s nucleus in the spinal grey matter of the lumbar cord after SCI [31]. In addition to neuronal protection, BDNF can enhance regeneration and sprouting of injured axons in the spinal cord or increase remyelination of injured axons. Based on the abovementioned findings, we could not rule out the possibility of a similar route of recovery of TPH2 KO vs. WT rats after spinal cord injury.

In the present study, we observed a clear difference in 5-HT content between spinal cord regions, with the highest levels in the lumbar areas, similar to previous reports [14,32]. It can be proposed that these results reflect the pivotal role of 5-HT in the regulation of the spinal reflexes responsible for hindlimb locomotor function. Several research laboratories have provided convincing evidence that 5-HT regulates the rhythm and coordination of movement through the central pattern generator (CPG). 5-HT has been recognized as a potent neuromodulator of CPG activity [33]. The CPG in the lumbar spinal cord is regulated both by supraspinal descending inputs that originate in the raphe nucleus and terminate in the intermediate gray and the ventral horn [34,35], as well as by sensory afferents [36]. Employing receptor-specific agonists and antagonists, which have varying binding affinities towards each of the 5-HT receptor subsets, researchers have identified the 5-HT1A, 5-HT2A/C [7], 5-HT3 [37] and 5-H7 receptors [38] as important players in the regulation of the spinal locomotor network, and they are able to modulate locomotion in experimental models of SCI when employed in conjunction with the activation of other monoaminergic systems [16]. Interestingly, the degree of locomotor recovery correlates with the level at which the transplantation of embryonic raphe neurons is performed. A better outcome is observed when the transplantation occurs at the T11 level, rather than T9, likely due to its closer proximity to the lumbar CPG and the resulting greater degree of serotonergic axon reinnervation of the lumbar cord [39]. Later, it was shown that greater efficacy for restoring locomotor activity could be obtained when transplantation after SCI was performed even closer to the CPG, at the level of lumbo-sacral spinal cord [40].

Local spinal circuits, including CPG networks, remain mostly intact below a spinal cord lesion but lose their descending inputs, both the specific commands running mostly over the cortico-, rubro-, vestibulo- and reticulospinal tracts and the unspecific modulatory bulbospinal inputs carried by the serotonin, dopamine and noradrenaline fibers [32]. CMAP parameters depend on the integrity of several anatomical structures, including the motoneurons of ventral horns and muscle fibers [41]. In the case of spinal cord stimulation, descending propriospinal tracts are also involved [42]. These neurons are found in the cervical enlargement projecting mainly caudally to the lumbo-sacral enlargement and thought to be important in mediating reflex control and in coordination during locomotion. Severe axon loss with incomplete reinnervation leads to a decrease in CMAP amplitude without a change in latency [43], which corresponds to our data shown in Figure 5. In various animal studies, it was shown that, regardless of the SCI model, there are necrotic and apoptotic neuronal deaths that occur, including propriospinal neurons, ventral motor neurons, Clarke’s column neurons and supraspinal neurons, in these areas [44]. Loss of innervation of skeletal muscle is accompanied by hormone level fluctuation, inflammation and oxidative stress damage resulting in skeletal muscle atrophy [45], which also affects CMAP amplitude [43]. In the present work, we observed significant deterioration of recovery of neuromuscular transmission post-injury in TPH2 KO rats in comparison to WT. This result is additional evidence of the importance of central 5-HT for the recovery of sensorimotor functions after SCI. The dynamics of recovery of serotonergic innervation of the ipsilateral and contralateral sides of the spinal cord after lateral hemisection in Wistar rats has already been shown previously by Leszczyńska et al. [46]. These authors also showed that recovery of locomotor functions is partly due to reinnervation of spinal cord circuits by serotonergic fibers. Therefore, a detailed study of changes in propriospinal pathways, motoneurons and muscles after hemisection in both groups in future studies is highly warranted.

In several other studies, TPH2 KO rodents displayed phenotypic changes such as, for instance, high levels of aggressive behavior [47,48,49,50,51], autistic-like behavior traits [52,53] and unexpectedly decreased anxiety-like behavior in comparison with WT controls [47,54]. Taken together, it can be concluded that TPH2 KO mice or rats exhibit significant neurochemical changes in the CNS and therefore could represent a relevant model for SCI studies.

## 4. Materials and Methods

### 4.1. Animals

All procedures were performed under the guidelines established by the European Community Council (Directive 2010/63/EU of 22 September 2010), and animal protocols were approved by the Ethics Committee of St. Petersburg State University, St. Petersburg, Russia (approval number 131-03-4 from 3 April 2023). The light cycle was 12:12 h with light starting at 8:00 a.m., and the room temperature was maintained at 22 C. Food and water were available ad libitum.

Female TPH2-deficient (TPH2 KO) (n = 6) and wild-type (WT) (n = 6) rats 13–16 months of age, on the Dark Agouti background [12], were used in the experiments. An independent cohort of female TPH2 KO (n = 6) and WT (n = 9) animals was used for the high-performance liquid chromatography (HPLC) analysis of serotonin levels in the spinal cord. To generate experimental WT and TPH2 KO animals, heterozygous (HET) TPH2 dams were bred with TPH2 HET males. Genotyping of animals was performed by PCR using primers TPH2_FW2: 5-ACC TGA GCC CAA GAG ACT TCC and TPH2_Rev2 following enzymatic digestion with Mln1 enzyme (Figure 1C).

### 4.2. Experimental Schedule

Prior to SCI, hindlimb motor function of all rats was tested in tapered beam walking test, ladder walking test and static rod test. After SCI, the same tests were performed for four consecutive weeks at the end of each week. Animal care and hindlimb scaling were performed daily (Figure 7A).

### 4.3. Surgery and Animal Care

SCI was generated by lateral hemisection at T8 spinal level [55] in each rat (Figure 7B–D). Surgical procedures were conducted in aseptic conditions under isoflurane anesthesia (1–2%; mixed with oxygen, at a flow rate of 0.8 L/min). The skin and muscles above T7–T9 vertebrae were opened, and the T9 vertebra was recognized by the shape of its spinous process, which is smaller and less sharp than in the neighbor T8 vertebra. Then, laminectomy was performed in T8 vertebra; the T7 and T9 vertebrae were fixed in spinal clamps, and a left-side lateral hemisection was made with a microscalpel using half of the needle inserted into the spinal cord in its midline to guide the blade. Then, the back muscles and skin were sutured separately by Vicryl 4 and Ethilon 4, respectively. An analgesic (ketorolac, 1 mg/kg, s/c) was given every 12 h for 48 h to relieve any post-operative pain. An antibiotic (enrofloxacin, 5 mg/kg, s/c) was given daily for 7 days to prevent urinary infection. Post-operatively, the bladder was emptied manually once a day until the complete recovery of voluntary micturition.

### 4.4. Motor Function Assessment

#### 4.4.1. Tapered Beam Walking Test

The tapered beam walking (TBW) test (Open Science, Krasnogorsk, Russia) was carried out using a beam of length 165 cm long placed 80 cm above the floor with a black box at the narrow end, which the rats were supposed to crawl into from the other end. The starting point was lit with a bright light, motivating the rats to move towards the black box. Three days prior to testing, the animals were trained in how to navigate the beam. The rats were video recorded as they were being tested, and the results were viewed afterwards in a frame-by-frame mode. A video camera (HC-V160, Panasonic, Osaka, Japan) was positioned at a slight ventral angle, so that both sides of the body and paw positions could be recorded simultaneously from a ventral view. Left (ipsilateral to the hemisection side) hindlimb function was assessed by counting the number of errors (missteps or slips from the top beam) and the total number of steps made. The average of three trials was calculated. The severity of sensorimotor deficit was expressed as percentage using the following formula: (errors/total number of steps) × 100%.

#### 4.4.2. Ladder Walking Test

The horizontal ladder rung walking (LW) test apparatus [56] consisted of stainless-steel rungs (3 mm in diameter) inserted 5 cm above the table surface between clear polycarbonate walls. The width of the alley was adjusted to the size of the animal so that it was about 1 cm wider than the rat to prevent it from turning around. In the present study, we used a regular pattern of rungs with a distance of 1 cm between rungs. For several days prior to testing, the animals were trained to cross the ladder from a neutral platform to reach their home cage. The test was repeated several times, and the total number of steps (approximately 15–20), missteps and slips were counted for each limb. Missteps were defined as complete faults when the limb fell between the rungs, causing balance and body posture disturbance. All testing sessions were video recorded. The degree of sensorimotor deficit was calculated using the following formula: ((missteps + 0.5 × slips)/total number of steps) × 100%.

#### 4.4.3. Static Rod

Along with motor coordination, balance capability was assessed in the static rod test [57]. The test apparatus consisted of a 100 cm-long and 28 mm-wide rod placed 80 cm above the floor, with one end fixed under the goal box and the other one hanging freely in the air. The fixed end had a mark 10 cm from the target box (“finish point”). Before testing, animals were trained to walk the rod towards the target box. During the three testing sessions, we averaged the total number of missteps. The degree of sensorimotor deficit was calculated similarly to the ladder walking test by the following formula: ((missteps + 0.5 × slips)/total number of steps) × 100%.

### 4.5. Scales for Assessment of Hindlimb Function Recovery

To assess spontaneous functional recovery of hindlimbs in rats after SCI, we also used a scoring system taking into account hindlimb muscle tone and capability of voluntary movement. The following point system was used for assessment of hindlimb muscle tone (response to passive flexion/extension): 5 points, strong response to passive flexion/extension; 4 points, good response; 3 points, moderate response; 2 points, weak response; 1 point, response is absent. For assessment of hindlimb voluntary movements, we used 1 (presence of voluntary movements) and 0 (voluntary movements are absent) scoring system. The degree of recovery was measured every day between 10 and 12 a.m. from the first day after SCI until the end of the study.

### 4.6. Terminal Electrophysiological Recording and Histological Examination

To assess neuromuscular conductance, we performed the terminal experiment and recorded compound muscle action potentials (CMAPs) in the gastrocnemius (GM) and tibialis (TA) muscles of both hindlimbs after stimulation of the sciatic nerve or lumbar region of the spinal cord 4 weeks after the lateral hemisection (n = 4 in both groups). Immediately after the terminal experiments performed under chloral hydrate (400 mg/kg b.w., i.p.) [58,59,60,61], the animals were deeply anesthetized with an overdose of tiletamine/zolazepam (Virbac, France, 100 mg/kg, i/m) and perfused transcardially with 0.9% NaCl (150) mL followed by 4% paraformaldehyde (300) mL in 0.1 M phosphate-buffered saline, pH 7.4. In the case of sciatic nerve stimulation, all experimental procedures were performed according to the protocol described by Pollari et al. [62] using 27G needles as electrodes. To selectively activate the GM and TA muscles, transcutaneous spinal cord stimulation was applied at L5–L6 vertebral level, where the dorsal and ventral roots of the segments L5–L6 containing motoneuronal pools of these muscles [63] were the most accessible for stimulation [64]. For that, the sacrum was recognized by a group of small spinous processes, immobile in relation to each other. The vertebra just rostral to the sacrum was counted as L6, and the patch (5 × 5 mm) sticky electrode was placed between the L6 and L5 spinous processes. Electrical stimulation (1 Hz frequency at stimulation intensities ranging from 1 mA to 12 mA in increments of 1 mA, 10 pulses for each stimulation amplitude, pulse duration of 0.1 ms) was applied to sciatic nerve or lumbar spinal cord using a Neuro-MEP electrical stimulator (Neurosoft, Ivanovo, Russia). To achieve supramaximal stimuli, the stimulation amplitude was elevated until the amplitude of the CMAP response reached a plateau. The maximum response was obtained with a further increase in the stimulation amplitude by 20%. Since the exact placement of the electrodes can affect the outcome value of the recording, we replaced the electrodes and repeated each measure three times using supramaximal stimuli to ensure that the largest response was obtained. CMAP maximum amplitude, latency and required stimulus were calculated for each trial.

A detailed dissection of vertebrae, roots and spinal cord was performed to determine the exact level and verify the spinal cord hemisection. The spinal cord was removed from the spine and stored in 20 and 30% sucrose until it sank. Blocks of segments T7–T9 were excised, including the epicenter of the hemisection and the surrounding segments, so that intact tissue rostrally and caudally to the lesion would be obtained. The blocks were cut on a freezing microtome into 50 mm transverse sections, mounted directly on the slides and stained with 0.1% toluidine blue (Sigma-Aldrich, St. Louis, MO, USA). For quantification of the lesion size, we calculated the length of the damaged tissue and the area of the spared tissue measured in frontal sections. The length of the lesion was calculated as a number of consequential frontal sections with detected lesion multiplied on section thickness. For the calculation of the spared tissue area, similarly to Asboth et al. [65], 3 sections from the serial frontal sections sequence were chosen: (1) the section without or with minimal tissue damage rostral to the hemisection, (2) the section with the minimal area of spared tissue and (3) the section without or with minimal tissue damage caudal to the hemisection. The areas of the undamaged tissue in all these sections were measured using Fiji software (Image J 1.54g), and the amount of the spared tissue was calculated taking the average area of undamaged sections as 100%.

### 4.7. HPLC Analysis of Serotonin Tissue Content

Animals were sacrificed by isoflurane overdose. The pieces of spinal cord at the cervical (C1–C3), thoracic (T7–T9) and lumbar (L2–L4) levels were dissected, weighed, snap frozen on dry ice and kept at −80 °C until further analysis by high-performance liquid chromatography with electrochemical detection (HPLC). Frozen tissue samples were homogenized in 710 µM ascorbic acid and 2.4% perchloric acid (Sigma-Aldrich, Steinheim, Germany), precipitated proteins were pelleted through centrifugation (20 min, 20,000× *g*, 4 °C) and the collected supernatant was analyzed for serotonin and its metabolite 5-hydroxyindoleacetic acid (5-HIAA) levels using high-sensitive reversed-phase HPLC with fluorometric detection, as described previously [66].

### 4.8. Statistical Analysis

Statistical analysis of the data was performed using GraphPad Prism 7.00 software (GraphPad Software Inc., San Diego, CA, USA). Significant differences were identified using Mann–Whitney U-test (to compare TPH2 KO and WT groups) and one-way repeated measures ANOVA (to compare matched groups). No rats were excluded from the study and the final analysis. Researchers were blinded to the group allocation at all stages of the experiment and data analysis. Data in charts are presented as M ± SEM. The significance threshold was set at *p* < 0.05.

## 5. Conclusions

Our findings suggest that TPH2 KO rats have a decreased capability of recovering sensorimotor function after SCI. TPH2 KO rats may be a valuable model to study the serotonergic mechanisms of spinal networks’ functional reorganization, understanding of which could lead to the development of novel pharmacotherapeutic approaches for treatment of patients with SCI.

## Figures and Tables

**Figure 1 ijms-26-02761-f001:**
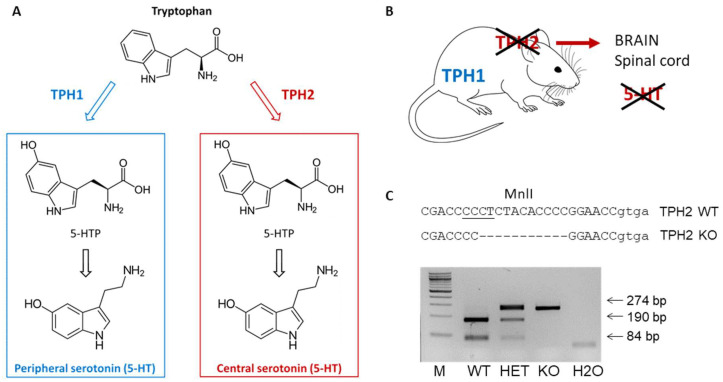
TPH2-deficient rats. (**A**) TPH2 is an enzyme specifically expressed in the serotonergic neurons and mediates serotonin synthesis in the CNS, whereas TPH1 is responsible for peripheral serotonin synthesis [13]. (**B**) A genetic deletion in exon 7 of the rat *Tph2* gene resulted in complete knockout of TPH2 and, consequently, a lack of serotonin in the brain [12]. (**C**) To genotype offspring, genomic DNA from ear biopsies was amplified with primers TPH2_FW2: 5-ACC TGA GCC CAA GAG ACT TCC and TPH2_Rev2: 5′-GCT ACG CTA TCA AAG GCC CG, and the resulting 274 bp-long PCR fragment was digested with the restriction nuclease Mnl1, which exclusively cuts the PCR products of the WT allele generating 190- and 84-bp long DNA fragments. 5-HT: 5-hydroxytryptamine; 5-HTP: 5-hydroxytryptophan; TPH: tryptophan hydroxylase. The MnlI restriction site is underlined.

**Figure 2 ijms-26-02761-f002:**
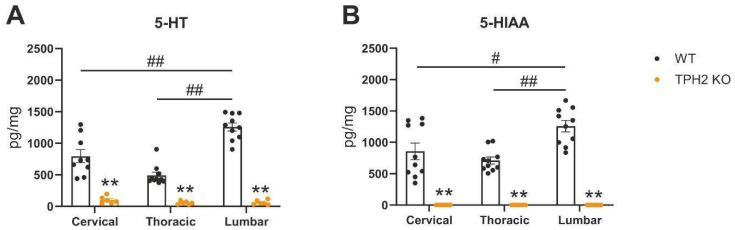
HPLC analysis of serotonin levels in the spinal cord. Serotonin (**A**) and its degradation product, 5-hydroxyindoleacetic acid (5-HIAA) (**B**), tissue contents were measured at cervical (C1–C3), thoracic (T7–T9), and lumbar (L2–L4) levels of spinal cord. ** *p* < 0.01 TPH2 KO vs. WT for the same area by Mann–Whitney U-test; # *p* < 0.05, ## *p* < 0.01—between areas for the same genotype by one-way repeated measures ANOVA followed by Tukey’s multiple comparisons test.

**Figure 3 ijms-26-02761-f003:**
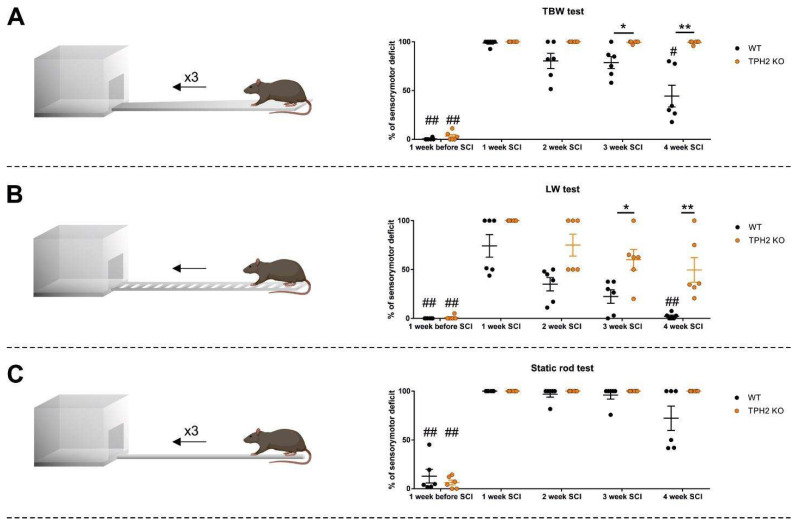
The degree of sensorimotor deficit in WT and TPH2 KO rats in TBW (**A**), LW (**B**) and static rod (**C**) tests before and after SCI. Each point represents an average result of trials for each rat. * *p* < 0.05, ** *p* < 0.01 between groups by two-tailed Mann–Whitney U-test, # *p* < 0.05, ## *p* < 0.01 vs. own values in the 1st week after the SCI by one-way repeated measures ANOVA followed by Tukey’s multiple comparisons test.

**Figure 4 ijms-26-02761-f004:**
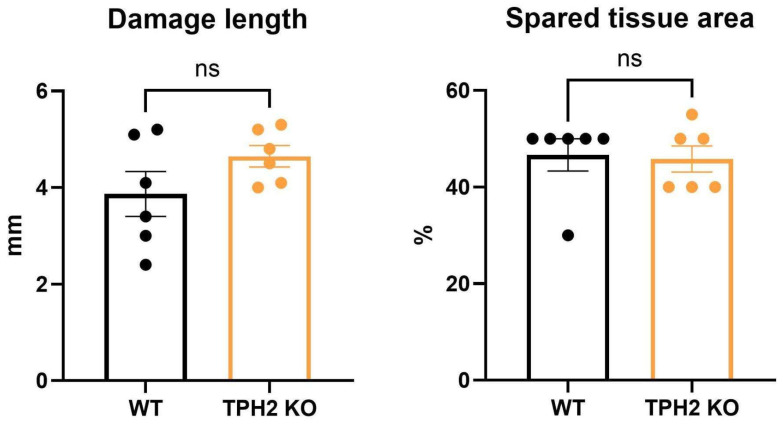
Comparison of the length of the lesion and spared tissue area between WT (n = 6) and KO (n = 6) groups. ns—non-significant differences between groups by two-tailed Mann–Whitney U-test.

**Figure 5 ijms-26-02761-f005:**
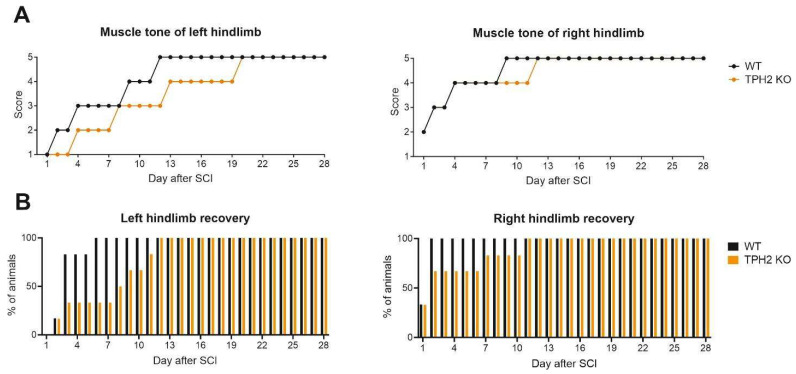
Recovery dynamics of the hindlimb muscle tone and voluntary movement in TPH2 KO and WT rats. (**A**) Muscle tone of left and right hindlimbs of WT (n = 6) and TPH2 KO (n = 6) rats after SCI. Points represent median results of each group. (**B**) Percentage of animals in the WT (n = 6) and TPH2 KO (n = 6) groups capable of ipsilateral (**left**) and contralateral (**right**) hindlimb voluntary movement after SCI.

**Figure 6 ijms-26-02761-f006:**
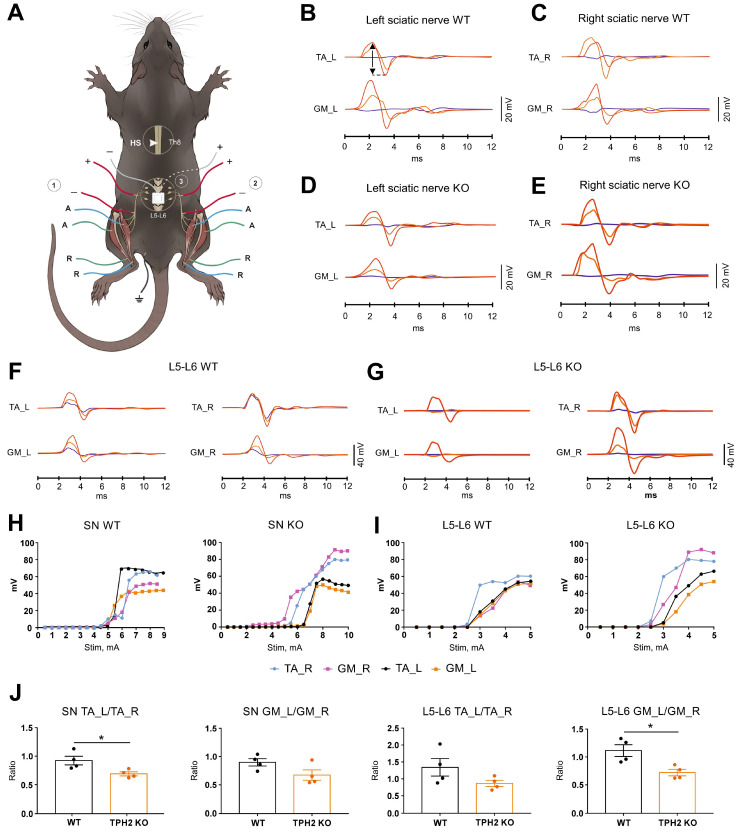
(**A**) Experimental setup for electrophysiological testing. EMG recording of GM and TA muscles during the stimulation of left sciatic nerve (1), right sciatic nerve (2) or L5–L6 lumbar level of the spinal cord (3). A, active electrode; R, reference electrode. (**B**–**G**). Examples of compound muscle action potentials in TA and GM muscles during the stimulation of sciatic nerve (SN) and L5–L6 spinal cord in WT and TPH2 KO rats. Blue, threshold responses; orange, submaximal responses; red, maximal responses. CMAP latency was measured to the first deviation of the potential from baseline. There were pronounced differences between the amplitudes of muscle responses on the left and right sides in the KO rat group. (**H**,**I**) Examples of the recruitment curves of TA and GM muscles during the stimulation of sciatic nerve (SN) and L5–L6 spinal cord. In the KO group, the thresholds of TA and GM muscles were lower from the left (ipsilateral) vs. right (contralateral) site. Also, similar stimulation amplitudes of SN or SC result in higher amplitudes of responses in TA_R and GM_R in comparison to TA_L and GM_R. (**J**) The ratio of left and right hindlimb maximum muscle responses (TA_L vs. TA_R and GM_L vs. GM_R) in TPH2 KO and WT groups during the stimulation of SN and L5–L6 spinal cord. * *p* < 0.05 between groups by Mann–Whitney U-test.

**Figure 7 ijms-26-02761-f007:**
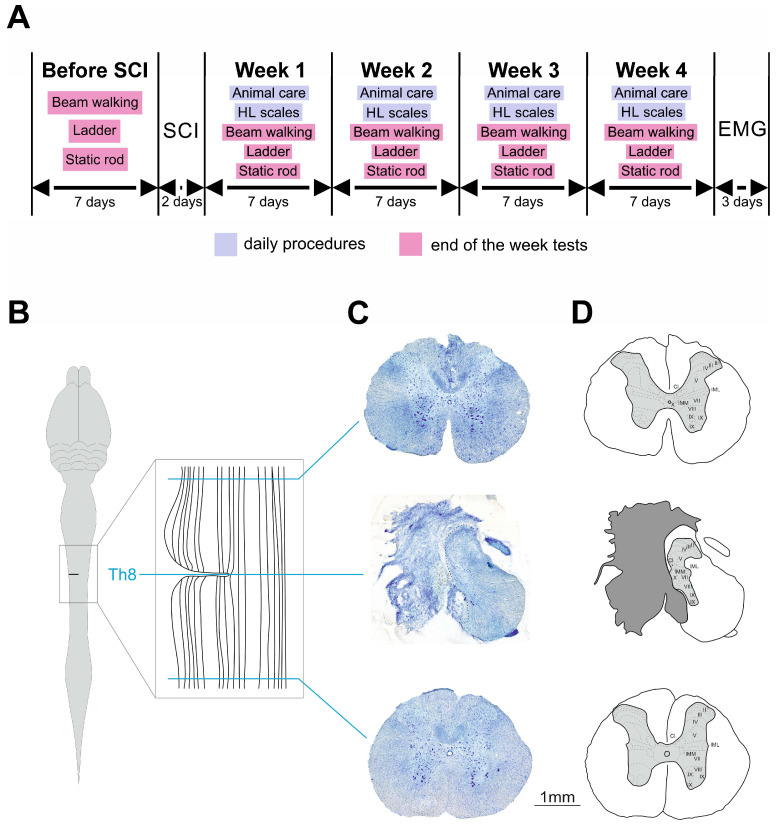
Experimental schedule (**A**) and lateral hemisection procedure (**B**–**D**). (**A**) Schedule of surgical procedures and behavioral and experimental test timeline. (**B**) Schematic illustration of lateral hemisection anatomical location. (**C**,**D**) Nissl-staining microphotograph (**C**) and the scheme of the anatomical infrastructure (**D**) above, below and on the level of SCI. SCI, spinal cord injury; HL, hindlimb; EMG, electrophysiological testing; I–X, Rexed laminae; IML, nucleus intermediolateralis; IMM, nucleus intermediomedialis; Cl, Clarke’s nucleus.

## Data Availability

The original contributions presented in this study are included in this article; further inquiries can be directed to the corresponding author.

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
