# Peer review of "Central Serotonin Deficiency Impairs Recovery of Sensorimotor Abilities After Spinal Cord Injury in Rats"

_ijms, 2025, doi:10.3390/ijms26062761_

Round 1
Reviewer 1 Report
Comments and Suggestions for Authors
The study investigates the role of central serotonin in sensorimotor recovery following spinal cord injury (SCI) using TPH2 knockout (KO) rats as a model. The results suggest that serotonin deficiency impairs functional recovery, highlighting the importance of serotonergic mechanisms in neurorehabilitation. The study is well-structured, presents clear experimental methodology, and utilizes multiple behavioral and electrophysiological assessments to support its conclusions.
However, the manuscript has several weaknesses that need to be addressed before it can be considered for publication. These include issues with clarity, methodological details, statistical interpretation, and overgeneralized conclusions. Below are specific points for revision.
1. The study only uses six rats per group, which is quite small, especially considering the variability in SCI recovery. While animal research is often limited by ethical concerns, statistical power calculations should be included to justify this small sample size.
2. Were outliers removed or handled in any way? If so, what criteria were used? The authors mention that no animals were excluded, but clarification on how they accounted for individual variability is needed.
3. The study emphasizes the role of serotonin in recovery but does not include direct histological evidence of serotonergic axon sprouting in the spinal cord post-injury.
4. It would be beneficial to stain for 5-HT fibers caudal to the lesion site to show whether serotonergic axons fail to regenerate in TPH2 KO rats. This would strengthen the claim that serotonergic deficiency impairs neuroplasticity.
5. The CMAP (compound muscle action potential) results suggest that TPH2 KO rats have impaired neuromuscular transmission post-injury. However: (1). Were these changes due to loss of descending drive, reduced synaptic connectivity, or muscle denervation? (2). A more thorough discussion on possible mechanisms underlying the CMAP differences is needed. (3). Were any spontaneous EMG recordings taken to assess baseline activity? This would help differentiate whether the issue is lack of voluntary motor output versus changes in spinal excitability.
6. The static rod test showed little recovery even in WT rats. Could it be that this test is too challenging and not the best measure of motor improvement?
7. Would it be beneficial to add grip strength tests or open-field locomotion analysis to better capture spontaneous recovery?
8. Did the authors blind the experimenters to the animal groups during behavioral scoring? If not, this introduces potential bias.
9. Some parts of the manuscript contain grammatical errors and awkward phrasing. For example: (1). "SCI induced severe motor deficiency in the left ipsilateral hind limb." Consider rewording to: "SCI caused severe motor impairments in the ipsilateral left hind limb." (2). "Serotonergic system seems to be crucial…" Consider rewording to: "The serotonergic system appears to be crucial…"
10. The Discussion section tends to be repetitive, particularly when reviewing prior studies on serotonin and recovery. A more concise discussion would improve readability.
11. Some of the electrophysiology plots are difficult to interpret. A clearer figure legend explaining recruitment curves and how latencies were measured would be useful.
Comments on the Quality of English LanguageSeveral sentences are awkwardly structured, overly complex, or grammatically incorrect.
Author Response
Comment 1. The study only uses six rats per group, which is quite small, especially considering the variability in SCI recovery. While animal research is often limited by ethical concerns, statistical power calculations should be included to justify this small sample size.
Response 1. Calculation of the sample size requires predicting the effect size of the outcome of interest. The effect size can usually be calculated using preliminary data observed in a smaller-scale study or in the literature for similar studies (doi: 10.3389/fmed.2023.1215927). Since there are no previous studies in THP2 KO rats after lateral hemisection, we can only suppose the optimal number of animals in both groups. Several authors used 3-7 rats in SCI studies (doi: 10.1093/brain/awr167, doi: 10.1089/neu.2018.6101, doi: 10.3390/brainsci15020191, doi: 10.1371/journal.pone.0294720, 10.7150/ijbs.84564). In general, our study presents preliminary data of recovery of TPH2 KO rats after lateral hemisection, which can be used as a basis for sample calculations in future studies. We agree that variability of SCI recovery may be a potential source of bias in our research, therefore, we added information considering damage site length and undamaged area in each animal used in the present study. Undoubtedly, the similarity of hemisections between groups does not guarantee the same rate of recovery (doi: 10.1016/j.heliyon.2019.e01324). However, it allows us to compare two groups in maximally similar experimental conditions.
Comment 2. Were outliers removed or handled in any way? If so, what criteria were used? The authors mention that no animals were excluded, but clarification on how they accounted for individual variability is needed.
Response 2. As we indicated earlier in the manuscript, no animals were excluded from the final analysis. We added information considering damage length and undamaged area in each animal used in the present study. There were no differences between groups in both analyzed parameters. Certainly, we observed individual variability between animals, for example, in fig.3A and 3C (unequal recovery in WT group on the 4th week after SCI).
Comments 3 and 4. The study emphasizes the role of serotonin in recovery but does not include direct histological evidence of serotonergic axon sprouting in the spinal cord post-injury. It would be beneficial to stain for 5-HT fibers caudal to the lesion site to show whether serotonergic axons fail to regenerate in TPH2 KO rats. This would strengthen the claim that serotonergic deficiency impairs neuroplasticity.
Responses 3 and 4. The dynamics of recovery of serotonergic innervation of the ipsilateral and contralateral sides of the spinal cord after lateral hemisection in Wistar rats has already been shown previously by Leszczyńska et al. (2015). These authors also showed that recovery of locomotor functions is partly due to reinnervation of spinal cord circuits by serotonergic fibers. In our follow up studies we are planning to assess direct histological evidence of serotonergic axon sprouting along with other aspects, such as changes in neurogenesis, sensory pathways etc. In the present work we focused mainly on the evidence of slower functional recovery in KO animals.
Comment 5. The CMAP (compound muscle action potential) results suggest that TPH2 KO rats have impaired neuromuscular transmission post-injury. However: (1). Were these changes due to loss of descending drive, reduced synaptic connectivity, or muscle denervation? (2). A more thorough discussion on possible mechanisms underlying the CMAP differences is needed. (3). Were any spontaneous EMG recordings taken to assess baseline activity? This would help differentiate whether the issue is lack of voluntary motor output versus changes in spinal excitability.
Response 5
This part was added to discussion:
Local spinal circuits, including CPG networks, remain mostly intact below a spinal cord lesion, but lose their descending inputs, both the specific commands running mostly over the cortico-, rubro-, vestibulo- and reticulospinal tracts, and the unspecific modulatory bulbospinal inputs carried by the serotonin, dopamine and noradrenaline fibres (Filli, 2011). СMAP parameters depend on the integrity of several anatomical structures including motoneurons of ventral horns and muscle fibers (Tavee, 2019). In the case of spinal cord stimulation descending propriospinal tracts are also involved (Laliberte, 2019). These neurons are found in the cervical enlargement projecting mainly caudally to the lumbosacral enlargement and thought to be important in mediating reflex control and in coordination during locomotion. Severe axon loss with incomplete reinnervation leads to decrease of CMAP amplitude without change of latency (Barkhaus, 2024), that corresponds to our data shown in Figure 5. In various animal studies it was shown that, regardless of the SCI model, necrotic and apoptotic neuronal deaths occur including propriospinal neurons, ventral motor neurons, Clarke’s column neurons, and supraspinal neurons in the areas (Hassannejad, 2018). Loss of innervation of skeletal muscle is accompanied by hormone level fluctuation, inflammation, and oxidative stress damage resulting in skeletal muscle atrophy (Xu, 2023) which also affects CMAP amplitude (Barkhaus, 2024). In the present work we observed significant deterioration of recovery of neuromuscular transmission post-injury in TPH2 KO rats in comparison to WT. This result is additional evidence of the importance of central 5-HT for the recovery of sensorimotor functions after SCI. Nevertheless, a detailed study of changes of propriospinal pathways, motoneurons and muscles after hemisection in both groups is highly warranted.
(3) We did not assess baseline activity due to the necessity of animal anesthesia. Several anesthetics, such as isoflurane or ketamine were shown to have neuroprotective activity. Others, such as chloral hydrate, are associated with high toxicity, and therefore can not be used except in a terminal experiment. CMAP recording is possible without anesthesia, but it requires chronic implantation of EMG electrodes which significantly increases the difficulty of the experiment.
Comment 6. The static rod test showed little recovery even in WT rats. Could it be that this test is too challenging and not the best measure of motor improvement?
Response 6. We completely agree with the reviewer. We believe that the static rod is a valuable and necessary sensorimotor test, however, in our case it would likely be more appropriate to use it at later stages after the hemisection.
Comment 7. Would it be beneficial to add grip strength tests or open-field locomotion analysis to better capture spontaneous recovery?
Response 7. It is a good point. We are planning to add the Open Field test to our future experiments but there are some limitations. For example, in this test the injured animal becomes accustomed to using the forelimbs, and misinterpretation of the results is possible. Therefore, additional analysis of hindlimb kinematic using Motorater, Catwalk, or other systems is necessary.
Grip strength is a highly sensitive test to motor abilities after spinal cord or brain injuries, but its usefulness is limited to forelimb function assessment. If there is a necessity to estimate hindlimbs motor function, it is better to use kinematic analysis of wading through shallow water (doi: 10.1093/brain/awr167; doi: 10.1038/nmeth.1484).
Comment 8. Did the authors blind the experimenters to the animal groups during behavioral scoring? If not, this introduces potential bias.
Response 8. Yes, the experimenters were blinded during behavioral scoring (they knew only the animal identification numbers, but not the number distribution between groups).
Comment 9. Some parts of the manuscript contain grammatical errors and awkward phrasing. For example: (1). "SCI induced severe motor deficiency in the left ipsilateral hind limb." Consider rewording to: "SCI caused severe motor impairments in the ipsilateral left hind limb." (2). "Serotonergic system seems to be crucial…" Consider rewording to: "The serotonergic system appears to be crucial…"
Response 9. Thank you for your comment. We thoroughly revised the manuscript.
Comment 10. The Discussion section tends to be repetitive, particularly when reviewing prior studies on serotonin and recovery. A more concise discussion would improve readability.
Response 10. Thank you for your suggestion. Two sections have been added to the discussion. We believe this could improve readability of the manuscript. We suppose that shortening the review of prior studies on serotonin and recovery can lead to an underestimation of the importance of the 5-HT system and especially the TPH2 KO model in SCI research.
Comment 11. Some of the electrophysiology plots are difficult to interpret. A clearer figure legend explaining recruitment curves and how latencies were measured would be useful.
Response 11. We added necessary information to the Figure 5
Figure 5. A. Experimental setup for electrophysiological testing. EMG recording of GM and TA muscles during the stimulation of left sciatic nerve (1), right sciatic nerve (2) or L5-L6 lumbar level of the spinal cord (3). A, active electrode, R, reference electrode. B-G. Examples of compound muscle action potentials in TA and GM muscles during the stimulation of sciatic nerve (SN) and L5-L6 spinal cord in WT and TPH2 KO rats. Blue, threshold responses, orange, submaximal responses, red, maximal responses. The latency of the CMAP was measured to the first deviation of the potential from baseline. There were pronounced differences between the amplitudes of muscle responses on the left and right sides in the KO rat group. H-I. Examples of the recruitment curves of TA and GM muscles during the stimulation of sciatic nerve (SN) and L5-L6 spinal cord. In the KO group, the thresholds of TA and GM muscles were lower on the left (ipsilateral) vs right (contralateral) site. Also, similar stimulation amplitudes of SN or SC resulted in higher amplitudes of responses in TA_R and GM_R in comparison to TA_L and GM_R. K. The ratio of left and right hindlimbs maximum muscle responses (TA_L vs TA_R and GM_L vs GM_R) in TPH2 KO and WT groups during the stimulation of sciatic nerve (SN) and L5-L6 spinal cord. *p<0.05 between groups by Mann-Whitney U-test.
Reviewer 2 Report
Comments and Suggestions for Authors
This study investigated the role of the serotonergic system in neuronal sprouting and regeneration in the spinal cord. For this, they have used tryptophan hydroxylase 2 knockout (TPH2 25 KO) rats deficient in serotonin in the brain and spinal cord. The
sensorimotor recovery after spinal cord injury was compared with the wild-type rats.
TPH2 is an enzyme that is specifically expressed in serotonergic neurons. It mediates the synthesis of serotonin in the CNS. Knockout rats exhibit a lower potential to recover motor functions than wild-type rats, thus supporting the underlying hypothesis.
The spinal tissue analysis shows higher serotonin levels in wild-type animals than in knock-out animals. Can authors elaborate on why the levels are higher in the spinal tissue from the lumber region than in the cervical and thoracic regions?
Various tests following spinal cord injury show a reduced deficiency in the wild-type than in knock-out animals. Data shown in Figure 3 are normalized to percent, but is it possible to include raw data, perhaps as a supplemental table so that readers can see the differences between the wild-type and knock-out animals? The additional studies support the overall role of serotonin in sensorimotor recovery following the spinal cord. Can authors include data indicating that neurological sprouting in the spinal cord after injury is greater in wild-type rats than in knockout rats, possibly through immunohistochemical analysis of spinal cord sections at various time points post-injury? This may help rule out other physiological differences between wild-type and knockout rats, beyond just serotonin levels, that could influence the recovery process.
Author Response
Comment 1. The spinal tissue analysis shows higher serotonin levels in wild-type animals than in knock-out animals. Can authors elaborate on why the levels are higher in the spinal tissue from the lumber region than in the cervical and thoracic regions?
Response 1. This part was added to discussion:
In the present study we observed a clear difference in 5-HT content between spinal cord regions, with the highest levels in the lumbar areas, similar to previous reports (14, Filli at al.2011 10.1093/brain/awr167). It can be proposed that these results are a reflection of the pivotal role of 5-HT in the regulation of spinal reflexes responsible for hindlimb locomotor function. Several research laboratories have provided convincing evidence that 5HT regulates the rhythm and coordination of movement through the Central Pattern Generator (CPG). 5-HT has been recognised as a potent neuromodulator of CPG activity (Feraboli-Lohnherr et al., 1997). The CPG in the lumbar spinal cord is regulated both by supraspinal descending inputs that originate in the raphe nucleus and terminate in the intermediate gray and the ventral horn (Carlsson et al., 1963; Ballion et al. 2002) as well as by sensory afferents (Rossignol et. al., 1998). Employing receptor-specific agonists and antagonists, which have varying binding affinities towards each of the 5-HT receptor subsets, researchers have identified 5-HT1A, 5-HT2A/C (Courtine et al., 2009), 5-HT3 (Guetrin and Steuer, 2005), and 5-H7 receptors (Liu et al., 2009) as important players in the regulation of the spinal locomotor network and which are able to modulate locomotion in experimental models of SCI when employed in conjunction with the activation of other monoaminergic systems (Musienko, 2011). Interestingly, the degree of locomotor recovery correlates with the level at which the transplant of embryonic raphe neurons is performed. A better outcome is observed when the transplantation occurs at the T11 level, rather than T9, likely due to its closer proximity to the lumbar CPG and the resulting greater degree of serotonergic axon reinnervation of the lumbar cord (Gimenez y Ribotta, 1996). Later it was shown that greater efficacy for restoring locomotor activity could be obtained when transplantation after SCI was performed even closer to the CPG, at the level of lumbo-sacral spinal cord (Orsaletal., 2002).
Comment 2. Various tests following spinal cord injury show a reduced deficiency in the wild-type than in knock-out animals. Data shown in Figure 3 are normalized to percent, but is it possible to include raw data, perhaps as a supplemental table so that readers can see the differences between the wild-type and knock-out animals?
Response 2. Thank you for your comment. We added raw data from fig.3 as a supplementary file.
Comment 3. The additional studies support the overall role of serotonin in sensorimotor recovery following the spinal cord. Can authors include data indicating that neurological sprouting in the spinal cord after injury is greater in wild-type rats than in knockout rats, possibly through immunohistochemical analysis of spinal cord sections at various time points post-injury? This may help rule out other physiological differences between wild-type and knockout rats, beyond just serotonin levels, that could influence the recovery process.
Response 3. Thank you for your suggestion. We are going to evaluate direct histological evidence of serotonergic axon sprouting like other findings, for example, changes in neurogenesis, sensory pathways etc. in future studies. In the present work we focused mainly on the functional evidence of slower recovery in KO animals.
Round 2
Reviewer 1 Report
Comments and Suggestions for Authors
The authors have thoroughly addressed all of my previous concerns. With these well-considered revisions, I believe the manuscript is now suitable for acceptance.